

# The Tibetan Plateau Space-based Tropospheric Aerosol Climatology: 2007–2020

Honglin Pan[1,2], Jianping Huang[1]*, Jiming Li[1], Zhongwei Huang[1], Tian Zhou[1], Kanike Raghavendra Kumar[3]

[1]*Collaborative Innovation Center for Western Ecological Safety, College of Atmospheric Sciences, Lanzhou University, Lanzhou, 730000, China*

[2]*Institute of Desert Meteorology, China Meteorological Administration, National Observation and Research Station of Desert Meteorology, Taklimakan Desert of Xinjiang, Taklimakan Desert Meteorology Field Experiment Station of China Meteorological Administration, Xinjiang Key Laboratory of Desert Meteorology and Sandstorm, Key Laboratory of Tree-ring Physical and Chemical Research, China Meteorological Administration, Urumqi, 830002, Xinjiang, China*

[3] *Department of Engineering Physics, College of Engineering, Koneru Lakshmaiah Education Foundation, Vaddeswaram, Guntur 522302, Andhra Pradesh, India*

Correspondence: Jianping Huang (hjp@lzu.edu.cn)



**Abstract.** A comprehensive and robust dataset of tropospheric aerosol properties is
important for understanding the effects of aerosol-radiation feedback on the climate
system and reducing the uncertainties of climate models. The third pole of Earth
(Tibetan Plateau, TP) is highly challenging to obtain long-term in situ aerosol data due
to its harsh environmental conditions. Here, we provide more reliable the new vertical
aerosol index (AI) parameter from the spaceborne-based Lidar (CALIOP) of CALIPSO
over TP during 2007-2020 between daytime and nighttime to investigate the aerosol's
climatology. The calculated vertical AI was derived from the aerosol extinction
coefficient (EC), which was rigorously quality-checked and validation, strictly quality
checked, and validated for passive satellite sensors (MODIS) and ground-based LIDAR
measurements. Generally, all those facts demonstrate the agreement of the AI dataset
with the CALIOP and ground-based LIDAR. Besides, all the evidence shows that after
removing the low-reliability aerosol target signal, the optimized data can obtain the
aerosol characteristics with higher reliability. Our data set also reveals the patterns and
numbers of high-altitude vertical structure characteristics of the aerosol troposphere
over the TP. Our dataset will help to update and makeup the observational aerosol data
in the TP. We encourage climate modeling groups to consider new analyses of the AI
vertical patterns, comparing the recovered datasets, with the potential to increase our
understanding of the aerosol-cloud-radiation-precipitation interaction and its climate
effects. Data described in this work are available at
https://data.tpdc.ac.cn/en/disallow/03fa38bc-25bd-46c5-b8ce-11b457f7d7fd
DOI:10.11888/Atmos.tpdc.300614. (Honglin Pan et al., 2023).

**Keywords:** Tibetan Plateau, Aerosol index vertical structure, Tropospheric aerosols,
Higher reliability, Aerosol climatology



## 1 Introduction

The three poles (i.e., the Arctic, Antarctic and Tibetan Plateau (TP)) have the highest mountains in the world and store more snow, ice and fresh water than any other place. The unique geographical location of the Antarctic, Arctic, and TP, as the unique ecological, climatic, and natural environmental changes, have crucial role in global and regional climate change. However, studies have found that these regions are susceptible to climate change and that their differences may also affect key feedback loops for global climate change and the sustainability of human societies. Unfortunately, our understanding of the three poles, particularly the relations between the regions, remains limited due to insufficient observation data. Currently, the collection of additional research data for these extreme environments is one of the major bottlenecks in facilitating comprehensive studies of these regions. Sufficient attention has been given to the polar regions and the TP in successive IPCC reports (IPCC, 2013;2021). The similarities between TP and the other two polar regions are their low temperatures, remote location, and large water storage capacity. On the other hand, TP has a more highly complex climate than the Arctic and Antarctic (where ice is the primary medium) and its land surface (including forests, grasslands, bare soil, lakes and glaciers) is more diverse. These differences make the transport and accumulation of pollutants in the TP region different from the other two polar regions.

The Tibetan Plateau (TP), is known as the "Third Pole" because it has the third largest ice mass on Earth, after the Antarctic and Arctic regions (Qiu, 2008). TP is also called the "Asia Water Towers", provides fresh water to 40% of the world's population due to its vast water reserves such as glaciers, lakes and rivers (Immerzeel et al., 2010). Furthermore, TP is the "Roof of the World", which covers an area of ~2.5 million km$^2$ at an average altitude of about 4,000 m a.s.l. (above sea level) and includes all of Tibet and parts of Qinghai, Gansu, Yunnan, and Sichuan in southwestern China, as well as parts of India, Nepal, Bhutan, and Pakistan (Nieberding et al., 2020). To the north of the TP region is situated by Taklamakan Desert (TD) (see Figure 1). This high altitude and specific topographic area effectively serve as a heat source during the spring and



summer months. This thermal structure helps the TP to function virtually as an "air
pump", attracting warm and humid air from the lower latitude oceans by suction (Yanai
et al., 1992; Wu and Zhang, 1998; Wu et al., 2007; Wu et al., 2012). Consequently,
large-scale mountains play a crucial role in shaping regional and even global weather
and climate through mechanical and thermodynamic effects and affect the global
energy-water cycle (Xu et al., 2008; Molnar et al., 2010; Boos and Kuang, 2010; Wu et
al., 2015). It is closely related to the survival of human beings in the world.

Climate projections are simulated responses of the climate system to future

emission or concentration scenarios of greenhouse gases (GHGs) and aerosols and are
generally calculated using climate models. The reasons for the gap between models and
observations may also be due to inadequate solar, volcanic, and aerosol forcing used in
the models, and in some models, may be due to an overestimation of the response to
increasing GHG and other anthropogenic forcing (the latter reason includes mainly the
role of aerosols). The most significant uncertainties in predicting future climate change
are related to uncertainties in the distribution and properties of aerosols and clouds,
their interactions, and limitations in the representation of aerosols and clouds in global
climate models (IPCC, 2021). The primary aerosol type over the TP is dust, and its
spatiotemporal pattern is primarily contributed to the Taklimakan Desert (Liu et
al.,2008; Chen et al., 2013;2022; Xu et al., 2015). Previously few studies of aerosol-
cloud-radiation-precipitation interaction have been conducted. For example, the dust
aerosols lifting over the TP reduce the radius of ice particles in the convective clouds
over the TP and prolong the cloud lifetime through the indirect radiation effect, which
can lead to the development of higher convective clouds. The dust-affected convective
clouds move further eastward under the action of westerly winds and merge with local
convective cloud masses, triggering heavy precipitation in the Yangtze River basin and
northern China downstream of the TP (Liu et al., JGR, 2019; Liu et al., NSR,2019).
However, the effect of aerosol on the atmospheric energy and water cycle remains
uncertain, mainly due to lacking long-term and accurate vertical aerosol optical
properties dataset over the TP. This can help better understand aerosol's impact on the
atmospheric heating rate and stabilization and the subsequent cloud-precipitation



process. Therefore, constructing a more long-term and reliable vertically dataset of
aerosol optical parameters can make up the observational facts for aerosol-related study
and provide a scientific basis for improving the global climate model simulation over
the TP.

Generally, the primary aerosol optical characteristic parameters (such as extinction

coefficient (EC), aerosol optical depth (AOD)) acquisition method is in situ
observations, which have high precision. However, in situ observations are restricted
by the distribution of observation stations over TP. Hence, the resulting data lack spatial
continuity, making it difficult to use to meet the objectives of growing regional
atmospheric environmental studies (Chen et al.,2022; Goldberg et al.,2019; Giles et al.,
2019). Satellite remote sensing (active and passive) is an effective tool for collecting
aerosol optical information (including the vertical structure and spatial distribution)
over a wide range of spatial scales, significantly offsetting the deficiencies of in situ
observations. Satellite remote sensing can tackle difficulties connected to insufficient
data and uneven geographical distributions to a certain extent (Chen et al., 2022; Wei
et al.,2021). While for aerosol products from CALIPSO, the presence of some low-
reliability aerosol target (LRAT) caused by cloud contamination, solar noise
contamination, especially in the daytime, and ground clutter among mostly aerosol
observations skews the distribution of the aerosol EC toward larger values, at least some
of which may be identified as aerosols and retained in the analysis, makes the presence
of some low confidence aerosol targets bias the distribution of aerosol extinction in
most aerosol observations. The distribution of the aerosol EC will show greater biased
values (Thomason and Vernier, 2013; Kovilakam et al., 2020; Pan et al., 2020; Kahn et
al., 2010), and then will further enhance the aerosol index (AI) value due to the
influence of radiation transfer interaction between clouds and the absorption layer,
which will not truly reflect the differences in aerosol physical properties (Guan et al.,
2008; Liu et al., 2019; Kim et al.,2018). Hence, gaining high confidence in EC helps us
analyze aerosol optical properties and better lead to numerous pertinent uses of EC data,
is essential for accurately characterizing the upper range of aerosol ECs that occur on
the TP.



The present study provides a dataset of monthly average vertical structure
characteristics of tropospheric high confidence aerosol optical properties including
extinction coefficient (EC), aerosol optical depth (AOD), Angstrom exponent (AE),
aerosol index (AI) between the daytime and nighttime over the TP and surrounding
areas. The data for the above-mentioned optical properties were retrieved based on the
space-borne Lidar CALIOP data (Cloud-Aerosol Lidar with Orthogonal Polarization)
from Cloud-Aerosol Lidar and Infrared Pathfinder Satellite Observation (CALIPSO)
satellite for the period 2007-2020. The main objective of this study is to calculate new
and high-confidence aerosol optical parameter of AI in the vertical distribution, by the
strict quality control and validation for passive satellite sensor (MODIS) and ground-
based LIDAR. Since AI is dependent on aerosol concentration, optical properties and
altitude of the aerosol layer, and AI is particularly sensitive to high-altitude aerosols,
which is used to indicate small particles (those that act as cloud condensation nuclei)
with a high weight (Guan et al., 2010; Buchard et al., 2015; Liu et al., 2019; Nakajima
et al., 2001). The comprehensive data set of aerosol properties utilized in the study is
of substantial importance for understanding the impact of aerosol on the ecosystem and
reducing the uncertainties of climate models.
The data set in this study can more effectively characterize the vertical structure
of aerosols while following standardized quality control methods to obtain higher
confidence in the aerosol vertical structural properties covariate data sets, and allow for
comparison and application to the study of climate models and other atmospheric
science related problems between our record and with different data sets. To ensure
meaningful confidence estimates for the constructed aerosol covariates over the TP, it
is necessary to apply carefully the following correction procedures and analytical
validation. The main steps to construct the dataset are grouped as follows: (1) Removing
the low-confidence aerosol extinction coefficient for 532nm and 1064nm caused by the
misclassification of cloud and other interferences (e.g., surface clutter, hygroscopicity
etc.). Based on this, an interquartile range (IQR) method (see section 2.2) is utilized to
discard the low confidence targets, and further obtain the monthly average aerosol EC
for day and night with higher confidence; (2) the pseudo-Angstrom exponent (hereafter



AE) is calculated using the EC at 532 and 1064nm with higher confidence; (3) obtaining
vertical AI by the product of the AOD (the vertical integral of EC) and AE. (4)
Validation for the constructed AI with: MODIS and in situ LIDAR measurements using
standardized frequency distributions.
**2 The construction of the data set**
**2.1 Study area**

Figure 1 depicts the geopotential height of the TP and its surrounding areas (27-

42° N,75-102° E, about 4,000 m a.s.l.), and schematic diagram of CALIPSO satellite
ground track over the TP in other months. The role of the "heat-driving air pump" of
the TP provides abundant water vapor for the formation of clouds (Luo et al., 1984;
Liou et al.,1986). Furthermore, the TP environment is greatly affected by natural and
anthropogenic aerosols from the surrounding regions (Chen et al., 2013; Bucci et
al.,2014; Xu et al.,2015). The strong convection generated by the TP will promote
aerosols' vertical transport and increase aerosols' content in the troposphere and
stratosphere (Vernier et al., 2015; Liu et al., 2022). Aerosols also serve as cloud
condensation nuclei (CCN) or ice nuclei (IN), modifying cloud structure properties and
precipitation (Twomey et al.,1977). Hence, the TP has been called the pumping pump
of water vapor, the clouds incubator, and the sand dust transfer station. By delivering
water vapor, clouds, and dust, it regulates extreme weather and climate in the
downstream and surrounding areas. It can be seen that the TP plays a crucial role in the
impact and regulation of global and regional climate or environments (Luo et al.,1984;
Rossow et al., 1999; Wan et al.,2017; Liu et al., 2022).














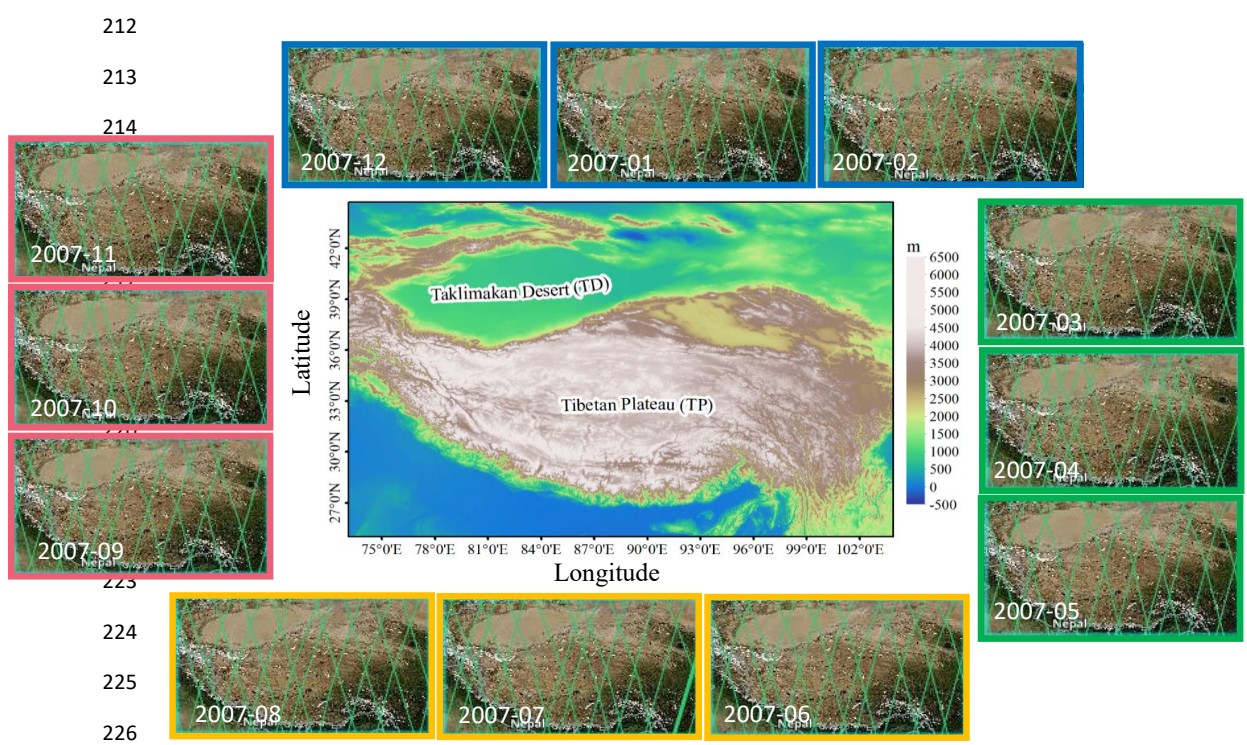

Figure. 1 The geopotential height of the TP and its surrounding areas (27-42°N,75-102°E),
schematic diagram of the transit of CALIPSO satellite orbits over the TP in other months (with 2007
as an example. March-May is spring, June-August is summer, September-November is autumn, and
December-February is winter).
**2.2 CALIPSO-CALIOP data and low-reliability aerosol target (LRAT) clearing**
**method**
CALIPSO (Cloud-Aerosol Lidar and Infrared Pathfinder Satellite Observations)
satellite was launched by NASA on 28 April 2006. The CALIOP (Cloud-Aerosol Lidar
with Orthogonal Polarization) onboard CALIPSO is the nadir-pointing dual-
wavelength polarization Lidar, which can provide the global and continuous
information on the vertical distribution of aerosols and clouds at 532 nm and 1064 nm
for daytime and nighttime. (Winker et al., 2007 and 2009). The CALIPSO-CALIOP
(version 4.20) level-2 aerosol profile product is selected in this study, with vertical and
horizontal resolutions of 60 m and 5 km, respectively. The used parameter includes



Extinction_Coefficient_532 and Extinction_Coefficient_1064 between daytime and
nighttime from 2007 to 2020. It should be noted that CALIOP observation data uses as
few instruments as necessary to complete the monthly aerosol climatology. We make
this decision to limit the impact of differences between instruments due to measurement
techniques and wavelength range as well as assess the general quality of the
instrument's data set.
The presence of some low-reliability aerosol target (LRAT) caused by cloud
contamination, solar noise contamination, especially in the daytime, and ground clutter
among mostly aerosol observations skews the distribution of the aerosol EC toward
larger values (Thomason and Vernier, 2013). Consequently, to eliminate the LRAT, a
statistical approach to identify LRAT, and extreme outliers is utilized based on the
interquartile range (IQR). IQR is a more conservative measure of the spread of
distribution than standard deviation (Iglewicz and Hoaglin, 1993). Note that this
technique is based on median statistics rather than the mean due to the skew distribution
of EC. In our implementation, we use daily data at each altitude (0.06 km) and latitude
(0.05°) bin from 2007-2020 to determine an EC frequency distribution for different
months. Besides, we used the lower quartile (Q1) and upper quartile (Q3) of the
underlying distribution to find IQR, defined as Q3-Q1, a good measure of the spread in
the data relative to the median. Here, an extreme outlier is defined as Q3 + (3.5×IQR),
and a more upper outlier (Q3+(1.5×IQR)) is used for comparison (Iglewicz and Hoaglin,
1993). Meanwhile, the extreme outlier threshold is used to clear LRAT-affected
observations from the data set, which is better and more effective at identifying outliers
in the density distribution (Kovilakam et al., 2020).
**2.3 AI Data processing**
According to the method described in section 2.2, the aerosol EC (observed at 532
nm and 1064 nm for daytime and nighttime) with higher reliability over the TP is
obtained. The monthly mean Ångström exponent (hereafter "pseudo-Ångström
exponent(AE)") between daytime and nighttime is derived to establish the 14-year
aerosol climatology (2007-2020) based on equation (1). The AE model for EC
wavelength dependence for 532 and 1064 nm is given by (Kovilakam et al., 2020):



$$EC_{-532[m,i,j]} = EC_{-1064[m,i,j]}\left(\frac{\lambda_{532}}{\lambda_{1064}}\right)^{AE[m,i,j]}$$

(1)

where EC_532 [m, i, j] and EC_1064 [m, i, j] are extinction coefficient at 532, and 1064
nm, respectively; AE [m,i,j] is the pseudo-Ångström exponent (Rieger et al.,2015;2019);
and the indices [m, i, j ] represent the month, latitude, and altitude respectively.
($\lambda_{532}/\lambda_{1064}$) represents the ratio of wavelengths at 532 and 1064 nm. The AE is gridded
to 0.05° latitude and 0.06 km altitude resolution. Further, the vertical distribution of
the new parameter AI is calculated according to equation (2). AI has been developed by
(Nakajima et al., 2001; Liu et al., 2019):
$AI_{[m,i,j]=} AOD_{[m,i,j]} \times AE_{[m,i,j]}$ , (2)
where $AI_{[m,i,j]}$ and $AOD_{[m,i,j]}$ are aerosol index and aerosol optical depth, respectively;
$AE_{[m, i, j]}$ is the pseudo-Ångström exponent; and [m, i, j] represent the month, latitude,
and altitude respectively. Note that to match the AE, AOD is also transformed into the
vertical distribution (not the column parameter). The monthly mean climatology of AI
is computed in altitude and latitude for 532 and 1064nm between daytime and nighttime.
**2.4 Aqua-MODIS data**
Like CALIPSO, Aqua is part of the A-Train constellation of satellites. Therefore,
MODIS (Moderate-resolution Imaging Spectroradiometer) onboard Aqua can achieve
near-simultaneous observations of clouds and aerosols with CALIPSO-CALIOP (less
than two minutes) (Winker et al., 2007; Hu et al., 2010). The Aqua satellite was
successfully launched on May 4th, 2002. Aqua is the afternoon star, passing through
the equator from south to north at around 13:30 local time. The observation data of 36
wavebands were obtained, with a maximum spatial resolution of 250 m and a scanning
width of 2330 km. MODIS is a passive imaging spectroradiometer, there are a total of
490 detectors distributed in 36 spectral bands, with full spectral coverage ranging from
0.4 microns (visible light) to 14.4 microns (thermal infrared). In this study, Level 3 data
(MYD08_M3) on a 1°×1° (longitude × latitude) gridded box is utilized. As shown in
Table 1, MODIS can provide 550 nm AOD and AE products. It is worth mentioning
that we chose this data because MODIS data is widely used and has certain reliability





299 in aerosol research. The parameters of AE and AOD from MODIS are also used to

300 calculate the AI, which is applied to evaluate the monthly mean climatology of AI from

301 CALIOP over TP (see Table 1).

302 Table.1 Comparison between MODIS and CALIOP existing data products (√ represents the existing

303 data products of the satellite, × represents data parameters that need further calculation in this

304 study).

| Detector/Satellite | Wavelength | Extinction Coefficient (EC) | Aerosol Optical Depth (AOD) | Angstrom Exponent (AE) | Aerosol Index (AI) |
|---|---|---|---|---|---|
| CALIOP/CALIPSO (active) | 532&1064nm | √ | √ | × | × |
| MODIS/Aqua (passive) | 550nm | × | √ | √ | × |
| | | | | verification | verification |

## 2.5 Ground-based LIDAR data

307 Besides, we use the ground-based LIDAR (Light Detection And Ranging) (38.967 °

308 N, 83.65 ° E, 1099.3m) detection data from the hinterland of the Taklimakan Desert

309 (TD) to verify the validity and accuracy of the low confidence aerosol removal method

310 and the AI calculated by CALIOP detection data. Multi-band Raman polarization

311 LIDAR (hereafter LIDAR) is mainly used for the detection of dust, aerosols, and clouds

312 particles in the atmosphere, which detection belongs to "Belt and Road" Lidar Network

313 from Lanzhou University, China (http://ciwes.lzu.edu.cn/), has an advantage with

314 calibrate or validate Satellite observation (see Figure 2). The primary technical

315 specifications of LIDAR are as given in Table 2. For the performance of this LIDAR

316 and the data inversion of aerosol related optical parameters, the authors advise the

317 readers to refer the research work of Zhang et al. (2022 and 2023).





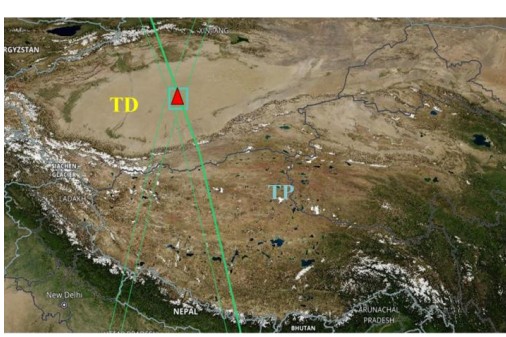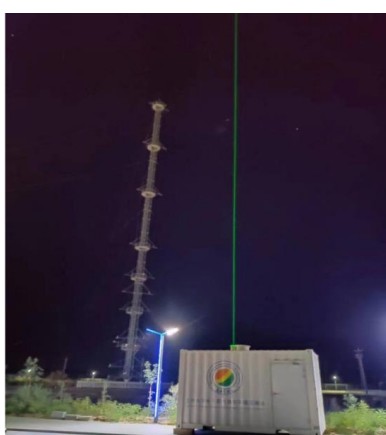


Figure 2. CALIPSO satellite orbit passes through the central area of the Taklimakan Desert
hinterland-left (the red triangle represents the observation coordinates of the ground-based LIDAR
- right (38.967 ° N, 83.65 ° E, 1099.3m), TD - Taklimakan Desert, TP - Qinghai Tibet Plateau)
(pictures from NASA'S Earth data (left) and photography(right)).
Table 2. Basic technical specifications of LIDAR from the hinterland of the Taklimakan Desert (TD).

| Detection range | Spatial resolution | Laser wavelength | Laser energy | Pulse frequency |
|---|---|---|---|---|
| 0~20km | 7.5m | 532nm/1064nm | 100mJ | 20Hz |


In this study, based on the Level_2 aerosol profile data product (extinction
coefficient, EC) for daytime and nighttime detected by CALIOP from 2007 to 2020,
the low-reliability aerosol target (LRAT) is screened and eliminated. The aerosol
characteristic data set with higher reliability over the TP is constructed, and the data set
is verified and compared with MODIS and ground-based LIDAR to test its
effectiveness and accuracy. Thus, the vertical structure of aerosol characteristics
climatology with higher reliability over the TP can be obtained, providing adequate
observation facts and a basis for the TP. All steps were implemented and was processed
as follows in figure 3.



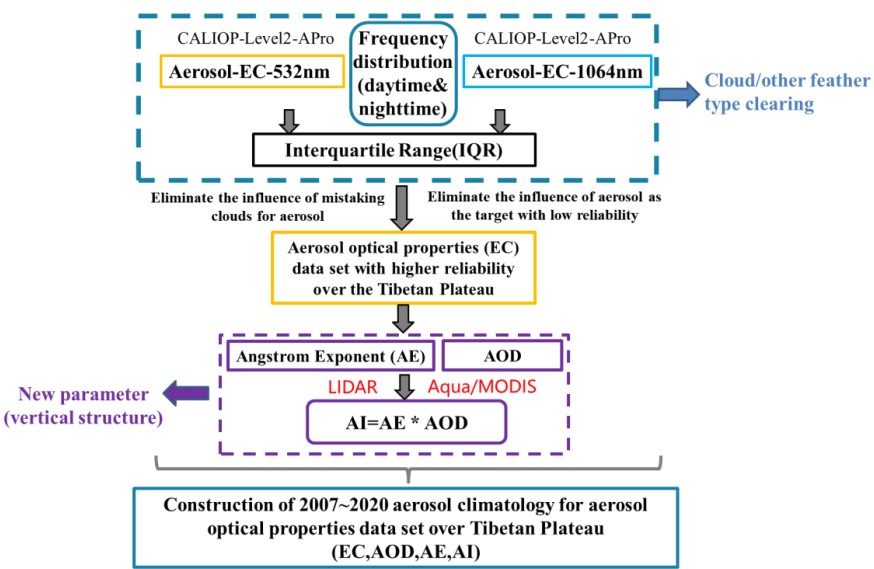


Figure 3. Flow chart of the aerosol characteristic data set construction and calculation process over

TP.

## 3 Results and analysis

### 3.1 Low-Reliability Aerosol Target (LRAT) screened and eliminate

In this section, we screened and eliminate LRAT for tropospheric aerosol extinction coefficient (EC) from the available CALIOP profile products over the TP, based on the statistical method (see Section 2.2). Figures 4 and 5 show the monthly frequency distribution of EC at 532 nm and 1064 nm in the daytime during 2007-2020 from January to December was detected by the CALIPSO-CALIOP troposphere within 12 km. While Figures 6 and 7 are for nighttime. Generally, Figures 4-7 demonstrate the non-normal distribution for EC. We found that the upper outlier appeared to remove many enhanced aerosol measurements, when more sand and dust events occurred in the surrounding areas and rose to the TP in spring and summer. In contrast, the extreme outlier was effectively identified in the frequency distribution. Therefore, the extreme outlier threshold used to clear LRAT observations from the CALIOP data set is necessary.

After the LRAT of screened and eliminate, we can directly compare these



measurements of the monthly climatology of data points and extreme outliers (2007 –
2020). We found that during the daytime for 532 nm and 1064 nm, the aerosol EC over
the TP is mainly concentrated between 0 and 0.2. The extreme outliers in July and
August are more significant than those in other months, which may be related to the
rising motion of the TP as a heat source in summer to trigger convection, resulting in
more ice clouds in the upper air, thus increasing the probability of misclassification the
cirrus anvil as an aerosol (Carrió et al., 2007; Kojima et al., 2004; Seifert et al., 2007).
Also, the aerosol data points (samples) is the largest in May and the smallest in
November over TP; Obviously, spring and summer are more than autumn and winter;
This is related to the frequent sand and dust activities in spring and summer around the
TP (such as Taklimakan Desert) and anthropogenic pollution (as mentioned earlier).

Similarly, during the nighttime for 532nm and 1064nm, the aerosol EC over the

TP is mainly concentrated between 0 and 0.1, and the extreme outliers in July and
August are still greater and more significant than those in other months. Still, it is
smaller than the daytime data set. The primary consideration is that the daytime solar
noise is considerable and the signal-to-noise ratio of LIDAR observation is low, which
further increases the probability that the aerosol EC presents skewed distribution; It can
be seen that the removal of LRAT from daytime data is more conducive to improving
the accuracy of data. Meanwhile, the aerosol data points are the largest in April and the
smallest in December over the TP. It can be seen that in April (spring), more aerosol
samples were lifted and transported to the TP. Numerous observations have shown
elevated dust plumes lofted into the free troposphere during spring, and air parcels
between 4 km and 7 km mainly originate from TD (Huang et al.,2008; Sasano,1996;
Liu et al.,2008; Zhou et al.,2002; Matsuki et al., 2003). It is the same as the daytime
with spring and summer being more than autumn and winter while there is one order of
magnitude larger than the data point in the day. It is not difficult to see that the main
reason is that the CALIOP is less sensitive during daytime than nighttime due to signal-
noise-ratio reduction by solar background illumination, which leads to weakly
scattering layers can be detected during nighttime while missed during daytime (Huang
et al., 2013; Liu et al.,2009).

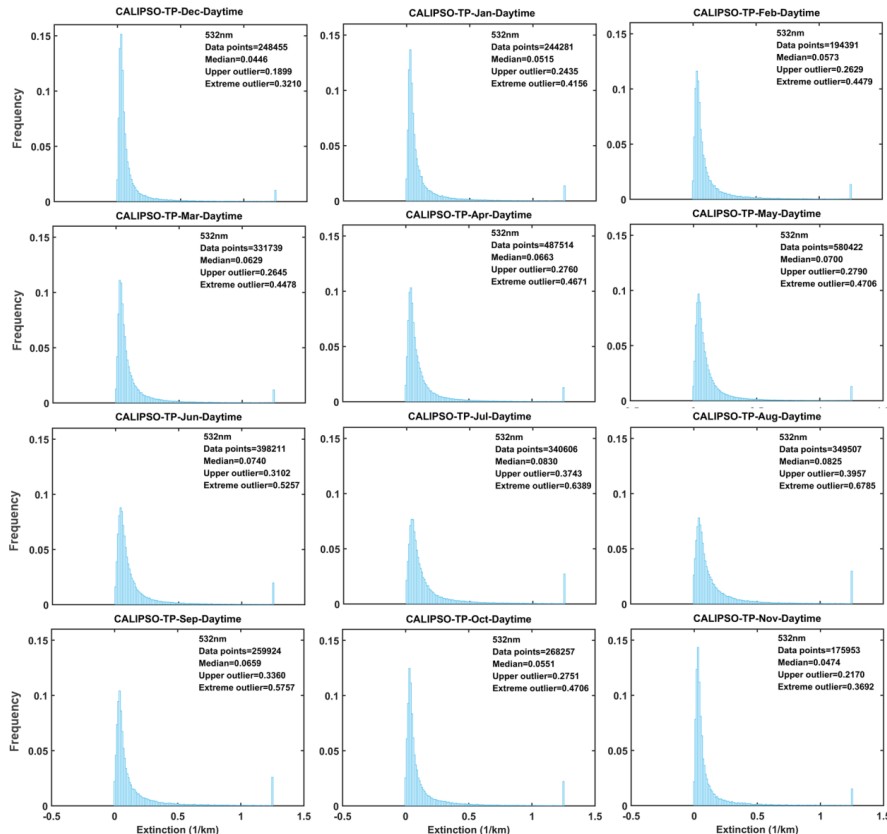


Figure 4. Monthly frequency distribution of aerosol extinction coefficient at 532nm over Tibet Plateau (TP) daytime during 2007~2020 from January to December (Panels 1st stands for Winter for Dec ~ Feb.; Panels 2nd stands for Spring for Mar ~ May; Panels 3rd stands for Summer for Jun ~ Aug; Panels 4th stands for Autumn for Sep ~ Nov). Frequency distribution is the number of events normalized to the maximum value. Upper outlier and extreme outlier and median also have been shown.







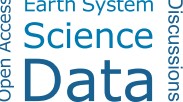



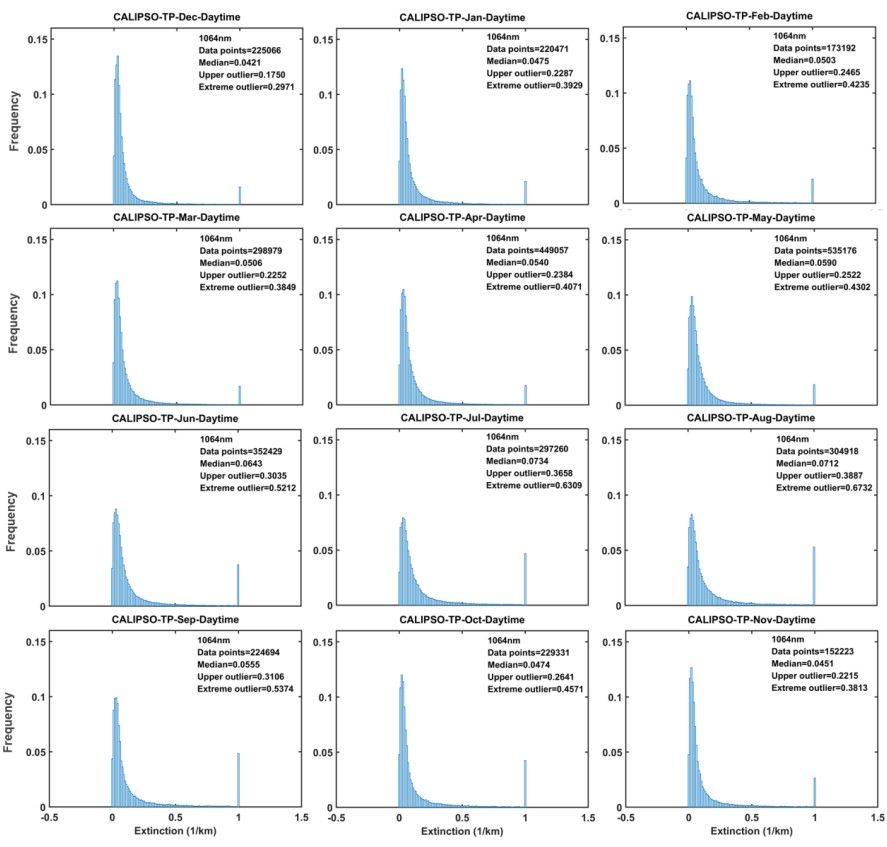

Figure 5. The same as in Figure 4 except for 1064nm.



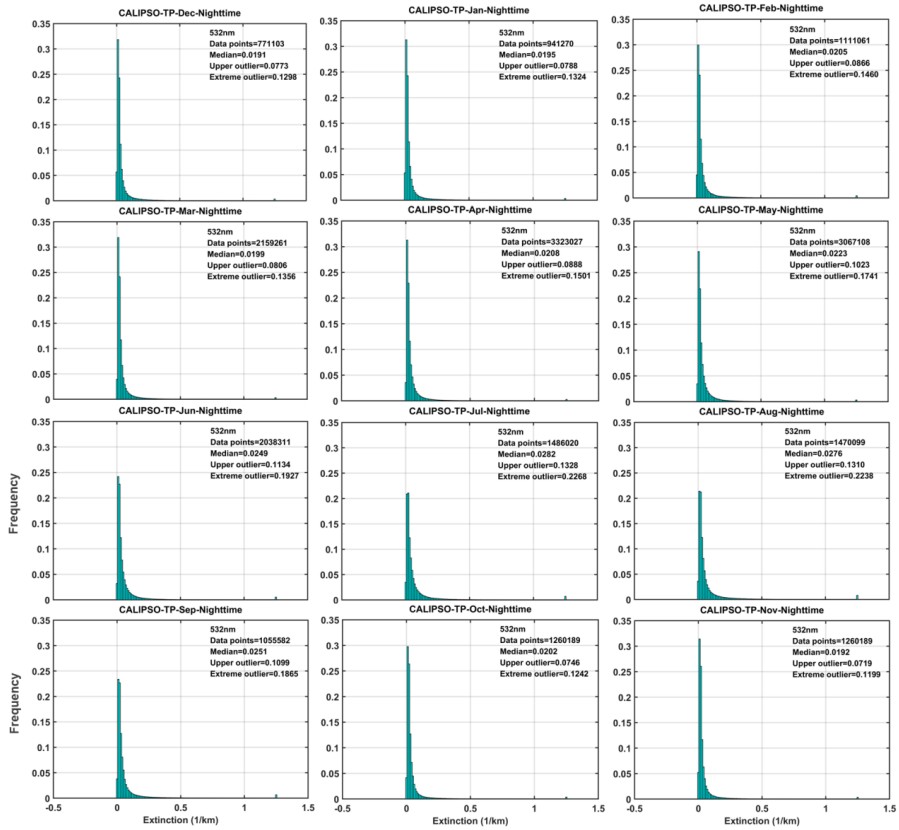

Figure 6. Frequency distribution of aerosol extinction coefficient at 532nm over Tibet Plateau (TP) nighttime during 2007-2020 from January to December (Panels 1st stands for Winter for December-February; Panels 2nd stands for Spring for March-May; Panels 3rd stands for Summer for June-August; Panels 4th stands for Autumn for September-November). Frequency distribution is shown as the number of events normalized to the maximum value. Upper outlier and extreme outlier, and median also have been shown.

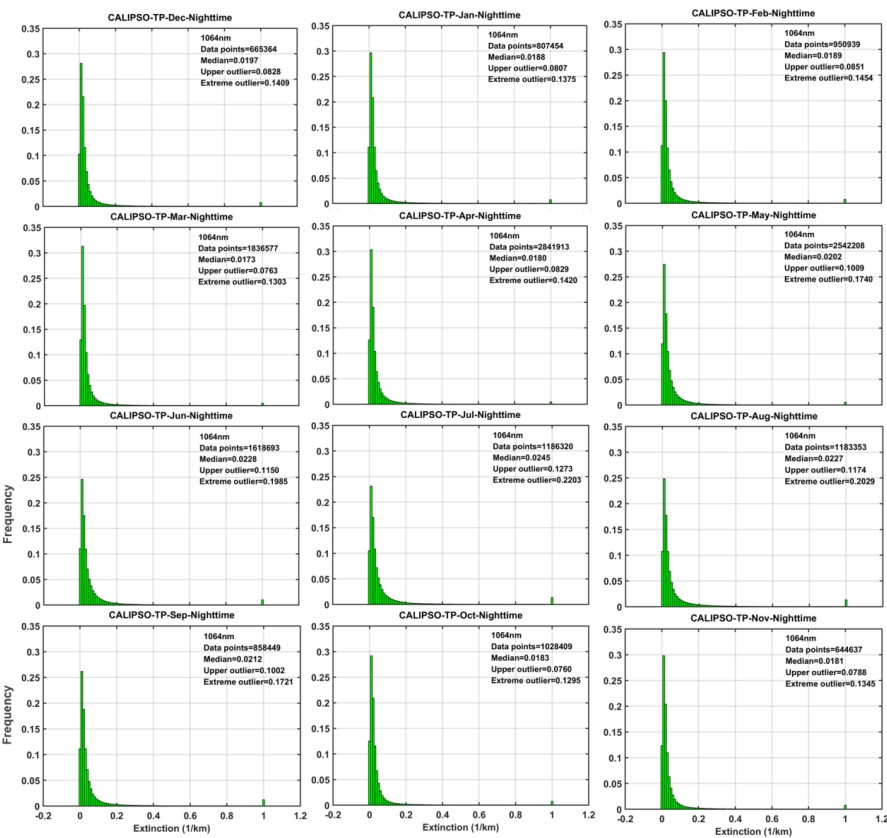

Figure 7. The same as in Figure 6 except for 1064nm.

### 3.2 Constructing vertical aerosol index (AI) for daytime and nighttime

Figures 8 and 9 show daytime altitude-latitude plots of the monthly climatology of the aerosol EC with 532 nm and 1064 nm before and after screen, respectively. The monthly mean climatology of the pseudo-Ångström exponent (AE) and Aerosol Index (AI) vertical structure is then computed (as shown in figure 10). We choose January, April, July and October to represent winter, spring, summer and autumn (same as below). Figures 8 and 9 show that extreme outliers in the troposphere over the TP have been eliminated, especially in the lower layer, where more obvious LRAT have been identified and eliminated. In the upper layer (more than 7 km), especially in April and July (i.e., spring and summer), weak cirrus signs may exist in the original aerosol signals and be eliminated. Compared with other seasons, the aerosol on the TP is widely and uniformly distributed in the troposphere in April, indicating that in general, more



aerosol loads are lifted over the TP in April. In figure 10, we compute values between
0 and -1 for much of the troposphere and occasionally are between 0 and 2 in the middle
troposphere (less than 8 km), which has similar results or pattern in Kovilakam's study
(Kovilakam et al., 2020). Note that the derived value for pseudo AE is without the
physical meaning, and it is simply a means to combine AOD to obtain AI of vertical
structure. Using this climatology of pseudo-AE values, we can effectively convert any
month of AI data to 532 nm and 1064 nm because the fixed AE is not necessarily
applicable to retrieving aerosol extinction in all months. Relevant research points out
that the accuracy has been improved, that is, using the corresponding AE index of each
month to correct the satellite data (Kovilakam et al., 2020).

Figure 10 also demonstrates the distribution characteristics of AI values at 532nm

and 1064nm in different seasons over the TP in the daytime. In all seasons, AI is mainly
distributed between -0.04 and 0.04. Still, the proportion between 0 and -0.02 is the
largest, indicating that the proportion of non-absorbability of tropospheric aerosols over
the TP is greater than that of absorbability of particles (AI, positively suggests the
existence of absorbent aerosols (dust, black carbon, etc.); A small or negative AI
suggests the presence of non-absorbable aerosols or clouds) (Hu et al.,2020; Guan et
al., 2010; Hammer et al., 2018). In cloud-free conditions, the highest and thickest
absorbing aerosols with the most prominent AI values, AI varies with aerosol layer
height, optical depth and single scattering albedo (Torres et al., 1998; 2007; Hsu et
al.,2004). In the four seasons, the distribution of aerosols in the north is broder than that
in the south; In spring, the rise height of aerosol is higher and the vertical distribution
range is more comprehensive; The elevation in summer is lower than that in the other
three seasons, but the aerosol species are more abundant, because there are many ranges
of AE values; However, the absorption aerosol below 7km in summer is less than that
in other three seasons.



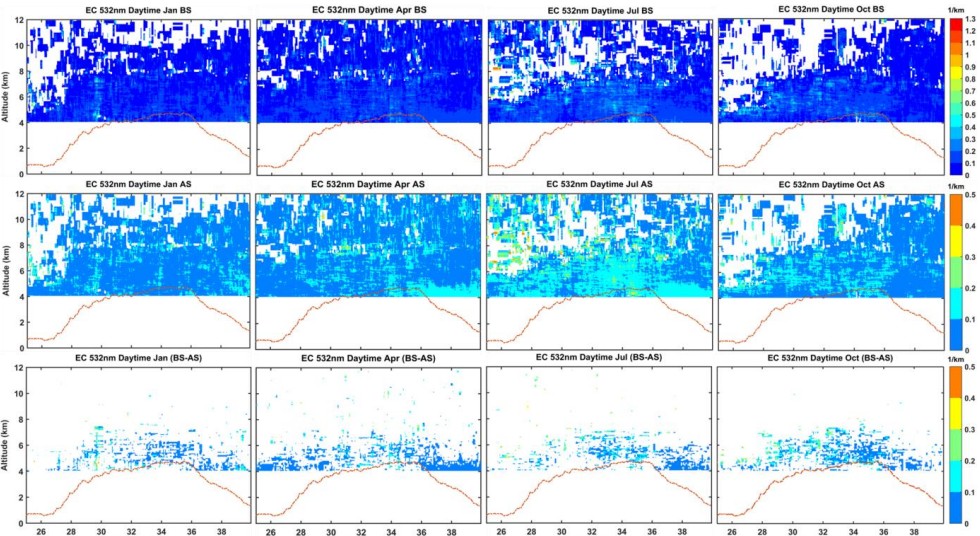

Figure 8. The monthly average comparison and difference of 532nm aerosol extinction coefficient before and after low-reliability aerosol target (LRAT) removal over Tibet Plateau (TP) daytime during 2007-2020. The reddish-brown dotted line denotes the surface. (BS: Before Screened, first line; AS: After Screened, second line; (BS-AS) means Before Screened minus After Screened, representing spatial lattice with screening and elimination, third line)

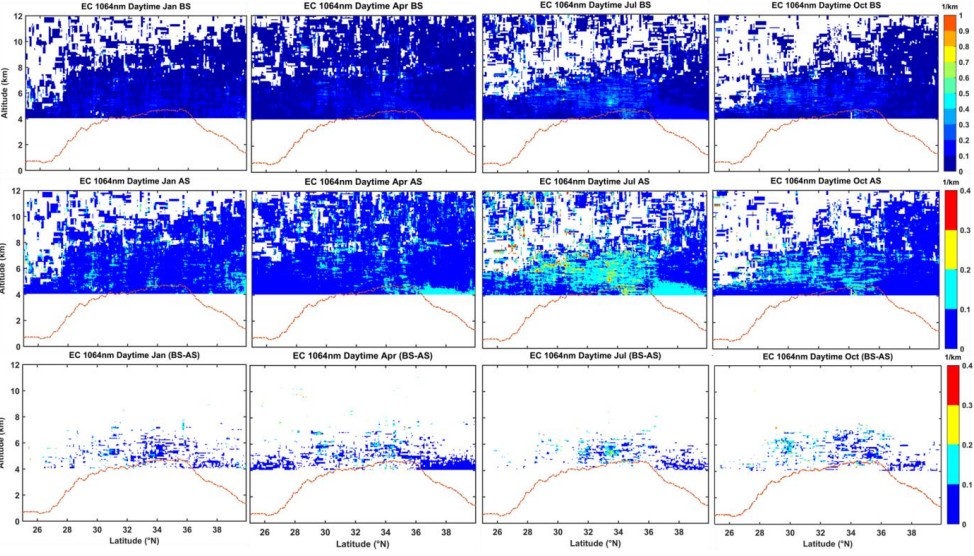

Figure 9. The same as in figure 8, but for 1064 nm.

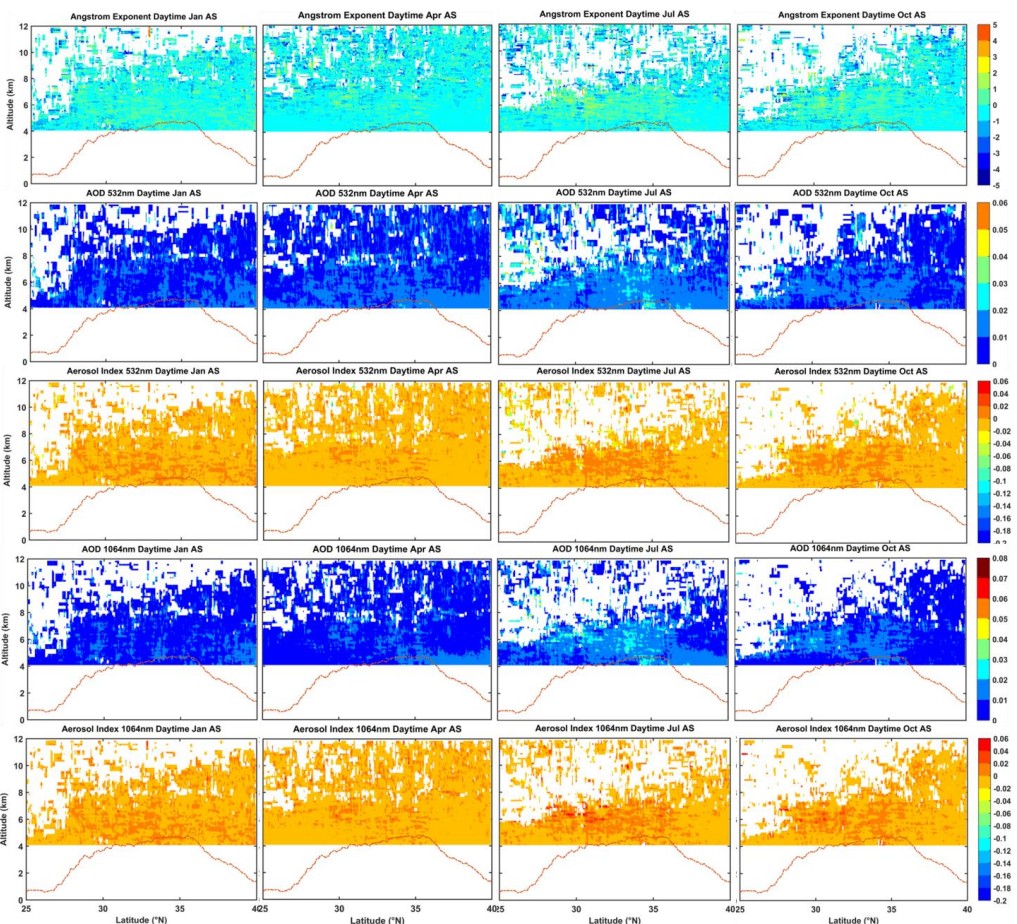

Figure 10. The monthly average construction of Angstrom Exponent (AE) and Aerosol Index (AI) of vertical structure for 532nm & 1064nm over Tibet Plateau (TP) daytime during 2007-2020.

Similarly, figure 11 includes the nighttime difference plots between the before-screened CALIOP 532nm EC and after-screened for different months during 2007-2020. The difference before and after screening is immense, especially at the height of more than 5 km in the southern region of the TP in July and October. We can see extreme outliers in the troposphere over the TP that have been recognized and eliminated. The EC detected at CALIOP 1064 nm shows a similar distribution characteristic as 532 nm, and also includes the different attributes before and after the screened and removal of LRAT (see as figure 12). In all seasons, AI is mainly distributed between -0.02 and 0.02. Still, the proportion between 0 and -0.02 is the largest in April and July (especially in

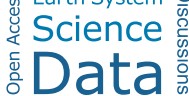

April, the non-absorbable aerosols dominate the 4-8 km high layer of the TP), indicating
the presence of non-absorbable aerosols. Meanwhile, AI above 8 km is mainly
concentrated at 0~0.02, indicating that the absorption aerosol is dominant. It is worth
noting that there is a large amount of absorbent aerosol over the TP in January (winter),
related to anthropogenic emissions of pollutants in winter and fossil fuel combustion
(such as black carbon and smoke). We note the pattern of AI is more or less consistent
with objective facts and phenomena.
Interestingly, compared with the daytime, the aerosol detected by CALIOP at night
can rise to a higher height and has a broader distribution range. It can be seen that
because the signal-to-noise ratio at night is higher than that in the daytime, CALIOP
can detect smaller particles, which is also why the quality and effectiveness of CALIOP
night detection data is better than that in the day. After a series of correction algorithms
and calculating relevant parameters, we have constructed the tropospheric AI
climatology dataset over the TP for 2007-2020.

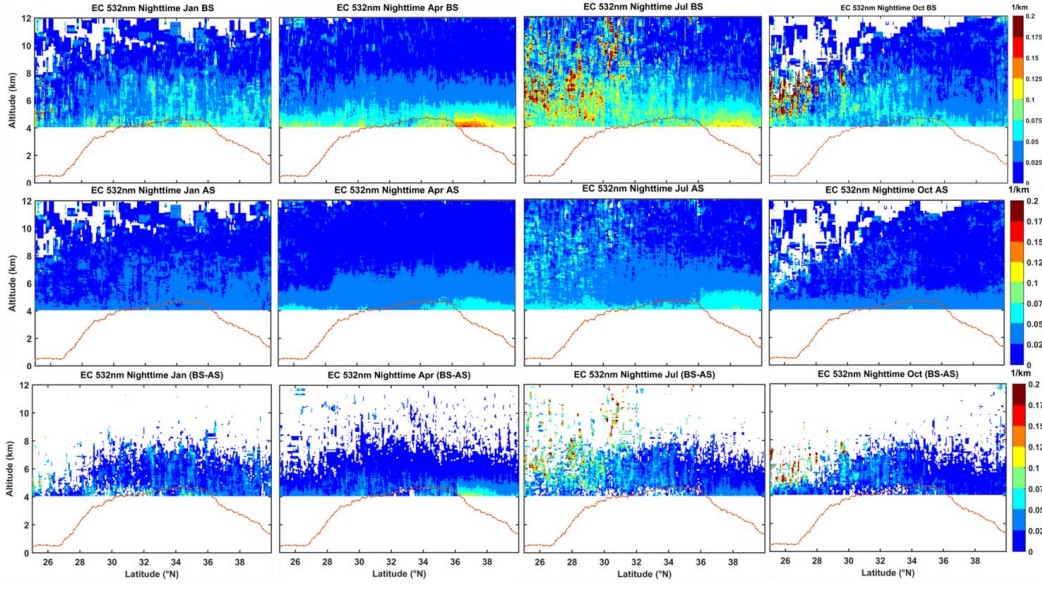


Figure 11. The same as in figure 8, but for nighttime.



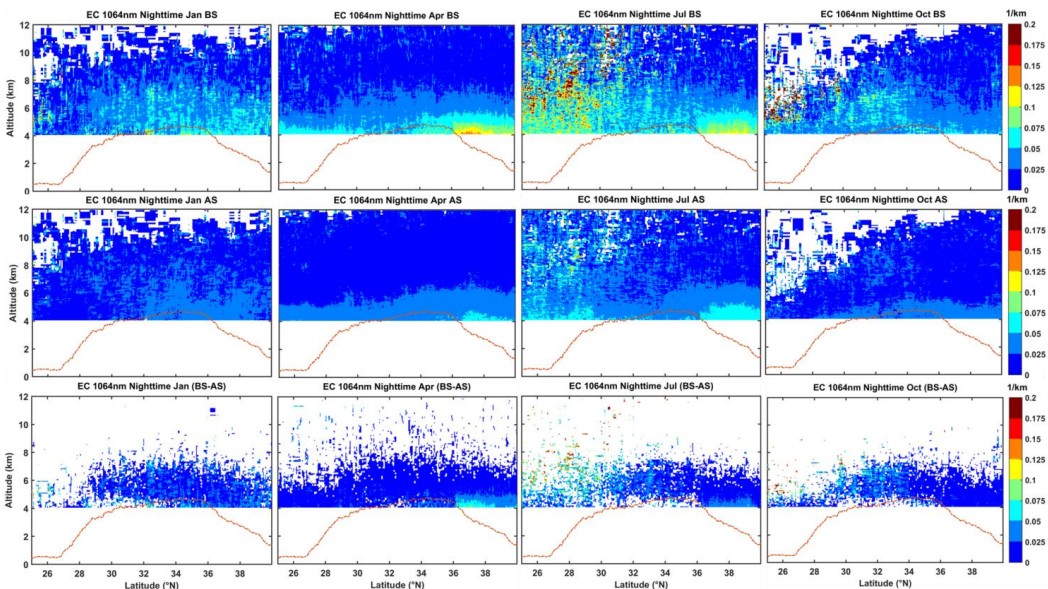

Figure 12. The same as in figure 11, but for 1064nm.

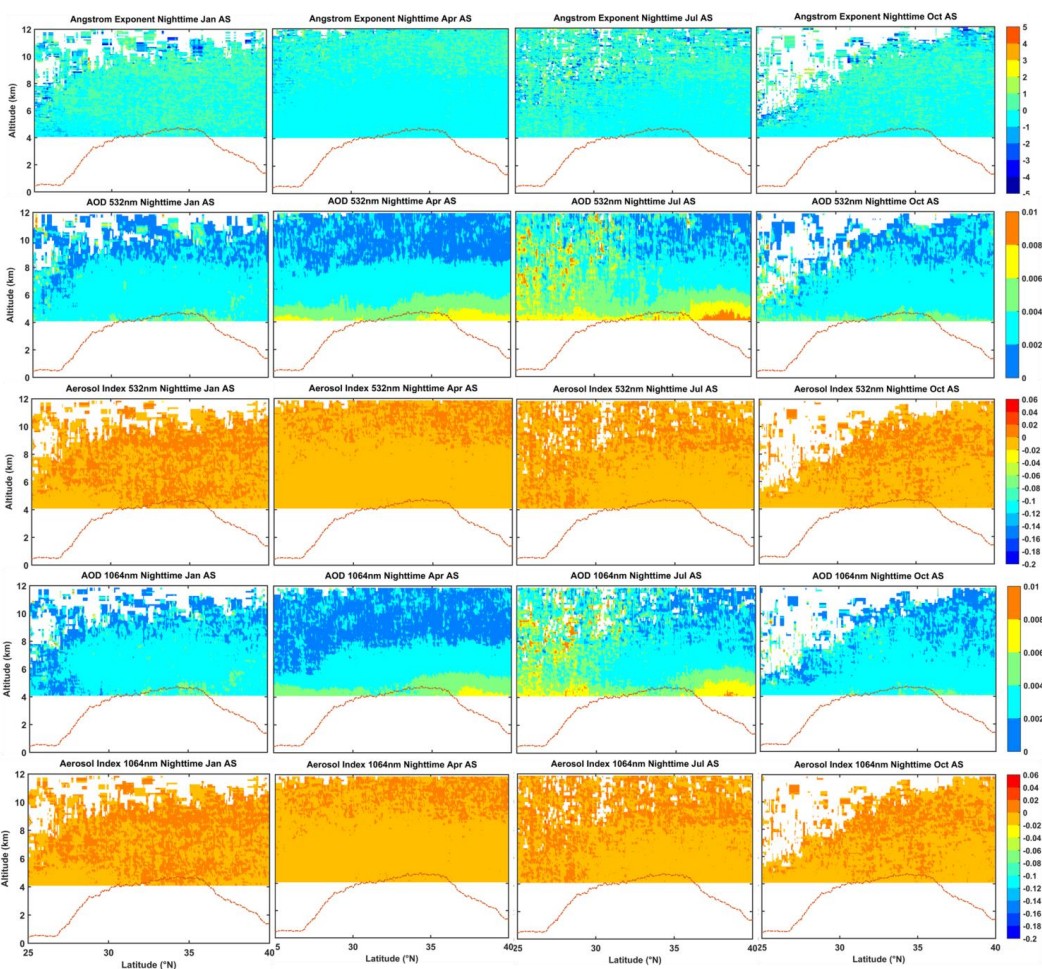

Figure 13. The same as in figure 10, but for nighttime.

### 3.3 Validation of the aerosol index (AI) dataset

#### 3.3.1 Comparisons with satellite Aqua-MODIS AI products

The multiyear monthly average spatial distributions of the AE and AOD from

MODIS have been shown in figure 14, and AI was also calculated (Figure 14). The

distribution of AE values over the TP in all seasons shows a decreasing trend from

southeast to northwest, indicating that the particles in the upper air of the southeast

region are dominated by small particles. In contrast, the particles in the upper air of the

northwest region are dominated by large particles, especially in April of spring, which

is related to the uplift and transmission of dust aerosol from the Taklimakan Desert to



the northern part the TP in spring. Additionally, we can see that the AE value of
Taklimakan Desert in the north of the TP in April and July in spring and summer is
smaller (as the source of the sand area, mainly dust aerosol), which is smaller than in
January and October in autumn and winter; AOD and AE showed opposite seasonal
variation distribution patterns. According to the spatial distribution pattern of AI
calculated from MODIS detection results (AE and AOD), it can be seen that the AI
value over the TP is mainly between 0 and 0.4. It shows that the primary existence is
an absorbent aerosol.

Figure 14 also compares the normalized frequency distribution of AI over the TP

exhibiting a significant difference in all seasons from MODIS and CALIOP between
BS and AS. It is evident that, in general, compared with the actual data results without
any processing, after removing the low-reliability aerosol target, the average AI value
of CALIOP is closer to the result of MODIS, and the normalized frequency distribution
pattern is closer to the same. Interestingly, the AI mean value and normalized frequency
distribution pattern of CALIOP in April (spring) after removing the LRAT are more
agreement and matched with the results of MODIS; In addition, the AI mean value and
normalized frequency distribution pattern of CALIOP in July (summer), and October
(autumn) is more consistent with the MODIS results, and both have apparent
improvement; The difference between the AI average value of CALIOP in January
(winter) and the result of MODIS is relatively more extensive, but the normalized
frequency distribution pattern is more consistent. This may be related to the type and
chemical composition of aerosol particles that rise over the TP in different seasons and
the atmospheric climate conditions unique to the topography of the TP. In brief, the
accuracy of aerosol parameters AI calculated after obtaining aerosol EC with higher
reliability has been dramatically improved (more or less), so even though not
completely accurate, this strategy is expected to reduce the inaccuracy of the computed
AI at least.

Meanwhile, it is proved that using extreme outliers as a limit to get more reliable

aerosol detection information is effective and reliable. It is important to note that the
550 nm wavelength range of MODIS belongs to the visible light range, and the data



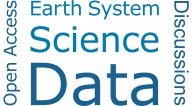

products provided at the satellite transit time are the daytime detection results.
Therefore, here we compare and verify the daytime detection results of CALIOP (532
nm) with MODIS results, which are consistent in time, close in detection wavelength,
comparable, and representative. In addition, the quality of CALIOP daytime detection
data is inferior to that at night, and the reliability and accuracy of the optimized data
are more effectively verified by comparison with the results of MODIS. Passive
techniques (i.e., MODIS) have the advantage of providing a 2-D distribution of AI over
a wide swath, during active strategies (i.e., CALIOP) with AI vertical structure. They
are complementary and have their advantages.





















Figure 14. Frequency test of AI calculated by MODIS-based aerosol AE and AOD over the Qinghai

Tibet Plateau and AI calculated by CALIPSO-based aerosol AE and AOD with high reliability for

daytime (BS: Before Screened, the fourth line; AS: After Screened, the fifth line).

*3.3.2 Performance evaluation based on in-situ Lidar observations*

To further verify the performance of the AI product derived from CALIOP over



the TP, we chose to use the ground-based LIDAR observation results in the center of
the Taklimakan Desert in the north of the TP to evaluate the effectiveness and accuracy
of the AI vertical structure of CALIOP.
To match the transit time of ground-based LIDAR observation and satellite
CALIOP observation, we extracted the EC (532 nm and 1064 nm) of ground-based
LIDAR during the daytime and nighttime to match the CALIOP adjacent observation
period, as shown in Figure 15 (observation case in TD on July 11, 2021, daytime: 03:00-
05:00, night: 14:00-16:00, China Beijing time, UTC+8). Considering the daytime
detection results of CALIOP for comparison and verification with MODIS in the above,
to further strengthen the inspection of CALIOP optimization results, we still choose the
daytime results of ground-based LIDAR detection for comparison and verification.
From Figure 15, it can also be seen that there are clouds or other LRAT in the daytime
high altitude in the ground-based LIDAR detection signal. This will be more beneficial
for us to check the validity and reliability of the results of the elimination of LRAT and
the calculated AI value.
Similarly, for ground-based LIDAR detection, we first reverse EC and use the IQR
method (see sec.2.2) to obtain extreme outliers and identify and eliminate the LRAT
(Figure 15). We can see that the LRAT (such as clouds and surface clutter etc.) are
effectively eliminated after the data optimization of 532nm and 1064nm detection
results EC. It is once again proved that it is effective and reliable to use extreme outliers
as a limit to obtain more reliable aerosol detection information.






Figure 15. Removal of low-reliability aerosol target signals detected by ground-based LIDAR in the
hinterland of Taklimakan Desert.

It is needed to be pointed out that the case of ground-based LIDAR detection on



July 11, 2021 is quite typical, but there is a significant deviation in satellite transit, and
this process cannot be well captured. To maximize and better match this process, we
take the ground-based LIDAR observation in the hinterland of the Taklimakan Desert
as the center (38.967 ° N, 83.65 ° E, 1099.3m), select 38.5~39.5 ° N and 83~84 ° E
range, extract the ECs observed by CALIOP transit in this range during the daytime
from 2007 to July 2020, and eliminate the LRAT. After averaging the optimized data,
further, calculate the AE value (as shown in Figure 16). Figure 16 depicts the detection
results of ground-based LIDAR and CALIOP optimal crossing point and the
comparison of calculated AI values. The AE values detected by ground-based LIDAR
and CALIOP are mainly distributed between - 1 and 1, and the proportion between - 1
and 0 is the largest. The aerosol can be raised to the height of 6 km, and the higher
concentration of aerosol is mainly concentrated below 2 km from the AOD vertical
layer, showing a decreasing trend with the increase of height; AI values are primarily
distributed between -0.02 and 0.02, and the average value and standard deviation trend
of AI change with height are also basically consistent. Generally, all those facts
demonstrate the agreement of the AI dataset with the CALIOP and ground-based
LIDAR. Besides, all the evidence shows that after removing the LRAT, the optimized
data can obtain aerosol characteristics with higher reliability.
Based on the monthly climatology AI product, we explored average vertical
structure change characteristics of AI over TP during 2007-2020 (as shown in figure
17). AI values in the daytime and at night over the Qinghai-Tibet Plateau mainly
fluctuate around 0, and the standard deviation increases with the increase of altitude.
The trend of AI changes with altitude is relatively consistent, and the standard deviation
below 6 km is slight, indicating that the dispersion of aerosol particles is small.
However, the fluctuation in the daytime is greater than that at night (the data quality at
night is better than that in the daytime). In general, the detection results of 532 nm and
1064 nm can achieve complementary observation. From the AI results at night, it can
be seen that the AI value of 532 nm over the whole troposphere over the TP is less than
0 in all months, indicating the existence of non-absorbable aerosols or clouds. We have
eliminated the interference of clouds, so there may only be non-absorbable aerosols. In



addition, when we look at 1064nm and the height above 8km, AI is positive, indicating
the existence of absorbent aerosols (dust and black carbon).
Figure 16. Comparative verification of AI of CALIPSO and ground-based LIDAR remote sensing
in Taklimakan Desert.



Figure 17. Monthly average vertical structure change characteristics of AI (mean & standard deviation) over TP during 2007-2020.



**4 Data availability**
Data described in this work are available at
https://data.tpdc.ac.cn/en/disallow/03fa38bc-25bd-46c5-b8ce-11b457f7d7fd
DOI:10.11888/Atmos.tpdc.300614. (Honglin Pan et al., 2023)

**5 Summary and outlook**
This present study is the first to report long-term, advanced-performance, high-
resolution, continuous and high-quality, monthly climatology aerosol AI vertical
structure from the CALIOP observation over TP which may be used to better
understand aerosol radiation forcing under the background of accelerated climate
change. Using the relationship developed when EC measurements are available, we
screened the entire EC record. We assembled a climatology of high-altitude aerosol
characteristics for daytime and nighttime from 2007 to 2020. In addition to providing a
monthly climatology AI data set for MODIS and ground-based LIDAR validation, our
data set also reveals the patterns and numbers of high-altitude vertical structure
characteristics of the aerosol troposphere over the TP.
To produce an accurate and higher reliability of AI values, we applied several
correction procedures and rigorously checked for data quality constraints during the
long observation period spanning almost 14 years (2007-2020). Nevertheless, some
uncertainties remain mainly due to technical constraints, as well as limited
documentation of the measurements. Even though not completely accurate, this strategy
is expected to at least reduce the inaccuracy of the computed characteristic value of
aerosol optical parameters. Following this initial work, we obtained vertical AI value
with higher reliability. This provides information about the vertical structures of aerosol
that could be used in climate models. The collection of more reliable and robust research
data sets of aerosol characteristics in these extreme environments is the key basis for
promoting comprehensive research on the energy balance of ground-atmosphere
radiation over the Tibetan Plateau and even the global region. We expect that this data
set will help some current and future research to simulate the climate change of the
monthly climatology. It will also help to update future data sets and study the interaction



of aerosol-cloud-precipitation, thus providing sufficient observation facts and basis.
**Author contributions.** HP led the reprocessing of the CALIOP, LIDAR, MODIS
measurements, data analysis and the preparation of the figures, with JH and JL both
contributing to design of the paper and progression of figures and text of the article. ZH
and TZ made the original LIDAR measurements. ZH provided the dataset and advice
on the re-processing of the LIDAR and CALIOP. KRK contributed to either
advising/co-ordinating the data recovery. All co-authors performed writing sections of
the paper, and/or reviewing drafts of the paper.

**Competing interests.** The authors declare that they have no conflict of interest.

**Acknowledgements.**
We are grateful to the CALIPSO (https://eosweb.larc.nasa.gov/), MODIS
(https://ladsweb.modaps.eosdis.nasa.gov/) instrument scientific teams at NASA for the
provision of satellite data, and "Belt and Road" Lidar Network from Lanzhou University,
China (http://ciwes.lzu.edu.cn/), which are available online and formed the central
database in the present work.

**Financial support.**
This work was financially Sponsored by the Natural Science Foundation of Xinjiang
Uygur Autonomous Region (Grant No. 2022D01B74), the Second Tibetan Plateau
Scientific Expedition and Research Program (STEP)(Grant No. 2019QZKK0602),
Scientific and Technological Innovation Team (Tianshan Innovation Team) project
(Grant No. 2022TSYCTD0007), National Natural Science Foundation of China (Grant
No. 42005074), Scientific Research and Operation Cost Project of Urumqi Institute of
Desert Meteorological, China Meteorological Administration (Grant No.
IDM2020003).

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
