# Peer review of "The Tibetan Plateau Space-based"

_Earth System Science Data, 2023_

## Referee Comment (RC2)

The Tibetan Plateau (TP) exerts crucial impact on in the changes of regional and even global weather and climate through mechanical and thermodynamic forcings and affect the global energy-water cycles. A comprehensive data of tropospheric aerosol properties over the TP is important for understanding the TP effects on climate and environment changes. The T is highly challenging to obtain long-term in situ aerosol data due to its harsh environmental conditions. Therefore, this study provides the new dataset of vertical aerosol index (AI) parameter during 2007-2020 to investigate the aerosol climatology over the TP, which is suitable for the ESSD publication. Here list some of my main comments:

In eq. 2, $AI_{[m,i,j]}$ and $AOD_{[m,i,j]}$ are aerosol index and aerosol optical depth, respectively; $AE[m, i, j]$ is the pseudo-Ångström exponent; and $[m, i, j]$ represent the month, latitude, and altitude respectively. Note that to match the AE, AOD is also transformed into the vertical distribution (not the column parameter) (lines 280-283). Please add the process transforming the column AOD into the vertical distribution with MODIS-AOD data.

The authors use the ground-based LIDAR (Light Detection And Ranging) detection data from the hinterland of the Taklimakan Desert (not from theTibetan Plateau) to verify the validity and accuracy of the low confidence aerosol removal method and the AI calculated by CALIOP detection data. Please add the discussions on the uncertainties in this study caused with distinct differences the Taklimakan Desert and the Tibetan Plateau.

the English language. Please make the substantial improvement on English language and usage in the manuscript. Below I list only a part of errors:

1) Line 38: more reliable the > the more reliable

2) Line 40: "between daytime and nighttime" should be for daytime and nighttime

3) Line 44: please clarify "all those facts"

4) Line 48: what is "aerosol troposphere" ?

5) Line 51: please modify "the recovered datasets"

6) Line 52: "the aerosol-cloud-radiation-precipitation interaction"?   There should be the aerosol-cloud interaction (ACI) aerosol-radiation interaction (ARI)..

7) Line 100: models > modeling

8) Lines 107-108: please modify "and its

9) Line 172: please correct "between our record and with different data sets"

10) Line108: spatiotemporal pattern is primarily contributed to the Taklimakan Desert".

---

## Author Comment (AC1)

Dear Editor,

Thank you very much for your help with the peer review of our manuscript. We are also very grateful to all the reviewers for their help to improve the quality of our manuscript. We have made a thorough revision of this manuscript according to the reviewers' recommendations. All modified content, including added or deleted parts, are marked with color (green) in the manuscript. Please see the following pages for a point-by-point response. We hope the revised manuscript is now suitable for publication in *Earth System Science Data*.

Kind regards,
Jianping Huang

This study aims to report an aerosol index dataset over the Tibetan plateau. The data is derived using CALIPSO extinction profiles at 532 and 1064nm and MODIS AOD. The authors claim that this dataset is useful in understanding aerosol composition and radiative forcing over the TP. However, I have two major concerns and confusions and think significant revision is needed.

Response: Thank you for your comments. We have carefully reviewed all the recommendations and made corresponding changes. Any inaccuracies in the text have been corrected per your suggestions. We hope that the revised manuscript will address all your concerns.

The scope of ESSD is to publish original dataset. While the AI data reported does not exist in literature, it is calculated from existing satellite products, i.e., CALIPSO extinction profile and MODIS AOD. The calculation also seems quite straight forward. Therefore, this dataset can hardly be considered original, but a reprocessing of existing data.

Response: Thank you for your comments. We mainly consider the following aspects:

Firstly, we remove the low-confidence aerosol extinction coefficient for 532nm and 1064nm caused by the misclassification of cloud and other interferences (e.g., surface clutter, hygroscopicity, etc.). Based on this, we use an observed relationship between CALIPSO extinction at 532nm and 1064nm, to derive an altitude-latitude-based monthly climatology of Ångström exponent to compute altitude-latitude monthly climatology of vertical AI at 532nm and 1064nm between daytime and nighttime from 2007 to 2020, resulting in a better agreement between MODIS and in situ LIDAR measurement. Moreover, the aerosol characteristic data we provide are vertical and highly reliable, which can effectively characterize aerosol concentration information at vertical heights. This was not present in previous work.

Secondly, we choose the CALIPSO-CALIOP (version 4.20) level-2 aerosol profile product in this study, with vertical and horizontal resolutions of 60 m and 5 km,

respectively. The used parameter includes Extinction_Coefficient_532 and Extinction_Coefficient_1064 (EC) between daytime and nighttime from 2007 to 2020. All work is carried out based on these data. The following is detailed data optimization processing, calculation process, as well as verification and evaluation process.

1) To eliminate low-reliability aerosol target, our technique is based on median statistics rather than the mean due to the skew distribution of EC. The presence of some low-reliable aerosol target caused by cloud contamination, solar noise contamination, especially in the daytime, and ground clutter among most aerosol observations skews the distribution of the aerosol EC toward larger values (Thomason and Vernier, 2013).

2) The pseudo Ångström exponent (hereafter AE) is calculated using the EC at 532 and 1064nm with higher confidence, effectively creating a 14-year monthly mean climatology of the AE based on measured values. We refer to the mechanism method developed by Kovilakam et al. (2020) to further extend the calculation of AI (Nakajima et al., 2001; Liu et al., 2019), obtaining vertical AI using the product of the AOD (the vertical integral of EC) and AE. All data are gridded to 0.05° latitude and 0.06km altitude resolution. The specific formulas have been explained and clarified in detail in the manuscript to provide readers with a relatively clearer understanding. It should be noted that we do not assume that the derived value for pseudo Ångström exponent has any physical meaning as it accounts for not only the actual behavior of aerosol, but also for potential deficiencies in both data sets, which is simply a means to further calculate AI. The AI allows for a more accurate quantification of aerosol column number concentration than AOD. We have unified definitions throughout the entire manuscript to eliminate inconsistencies or ambiguities.

3) Validation for the constructed AI with MODIS and in situ LIDAR measurements using standardized frequency distributions. Why do we choose MODIS to validate? MODIS (Moderate-resolution Imaging Spectroradiometer) onboard Aqua can achieve near-simultaneous observations of clouds and aerosols with CALIPSO-CALIOP (less than two minutes). There is a certain degree of synchronization in data detection. Besides, MODIS can provide 550 nm AOD and AE products, which is very close to the 532nm detection wavelength channel provided by CALIOP, and to some

extent, the verification results are representative. It is worth mentioning that we chose this data because MODIS data is widely used and has certain reliability in aerosol research. In addition, why do we choose LIDAR to validate in situ? Due to the lack of ground-based LIDAR detection data on the Qinghai Tibet Plateau, we use the ground-based LIDAR detection data from the hinterland of the Taklimakan Desert (TD) to verify the validity and accuracy of the low confidence aerosol removal method and the AI calculated by CALIOP detection data. This also plays a certain role in verifying and evaluating the results of satellite remote sensing data or the processing of new parameters.

Lastly, our processing and effective validation of satellite remote sensing data, as well as the construction of new parameters for aerosol vertical characteristic distribution, are all based on previous literature research results, not simple satellite data processing processes. For example, in our literature review, we found that Kovilakam et al. (2020) had also combined multi-source satellite remote sensing detection data for calculation, integration, comparative verification, etc., and constructed a dataset of aerosol characteristics for the long time series of stratospheric aerosols. Their research effectively characterized the climate change characteristics of aerosols, and relevant results were published in *Earth System Science Data* in 2020.

All in all, based on the Level_2 aerosol profile data product (extinction coefficient, EC) for daytime and nighttime detected by CALIOP from 2007 to 2020, we screened and eliminated the low-reliability aerosol target (LRAT). We also constructed an aerosol characteristic data set with higher reliability over the TP, verified the data set, and compared it with MODIS and ground-based LIDAR to test its effectiveness and accuracy. Thus, the vertical structure of aerosol characteristics climatology with higher reliability over the TP can be obtained, providing adequate observation facts and a basis for the TP.

Thomason, L. W. and Vernier, J.-P.: Improved SAGE II cloud/aerosol categorization and observations of the Asian tropopause aerosol layer: 1989–2005, Atmos. Chem. Phys., 13, 4605–4616, https://doi.org/10.5194/acp-13-4605-2013, 2013.

Kovilakam, M., Thomason, L. W., Ernest, N., Rieger, L., Bourassa, A., Millán, L.: The global space-based stratospheric aerosol climatology (version 2.0): 1979–2018, Earth System Science Data, 12(4), 2607-2634, 2020.

Nakajima, T., Higurashi, A., Kawamoto, K., Penner, J. E.: A possible correlation between satellite - derived cloud and aerosol microphysical parameters, Geophysical Research Letters, 28 (7), 1171-1174, https://doi.org/10.1029/2000GL012186, 2001.

Liu, Z., Kar, J., Zeng, S., Tackett, J., Vaughan, M., Avery, M., Pelon, J., Getzewich, B., Lee, K.P., Magill, B., Omar, A., Lucker, P., Trepte,C., Winker, D.: Discriminating between clouds and aerosols in the caliop version 4.1 data products, Atmos. Meas. Tech., 12, 703–734, 2019.

The definition, meaning and calculation of AI is quite confusing and not consistent throughout the manuscript. The authors define AI as AOD*AE. This type of AI is sometimes considered as a proxy of anthropogenic CCNs, or aerosol column number (e.g., Nakajima, 2001) and is often used in aerosol-cloud interaction studies. But later in the manuscript, the authors seem to refer AI as a representation of absorbing aerosols, e.g., lines 639-640, 642-643. Note the absorbing AI is defined in a different way as the radiance contrast between two channels of the Rayleigh atmosphere and polluted atmosphere (Torres et al., 1998). Even if the authors want to calculate the CCN proxy AI, I don't understand the logic of using column AOD but layered AE? What is the rationale for this? Shouldn't the AI also correspond to a specific aerosol loading to represent anthropogenic aerosols? Please unify the AI definition and clarify its physical meaning.

Torres, O., Bhartia, P.K., Herman, J.R., Ahmad, Z. and Gleason, J., 1998. Derivation of aerosol properties from satellite measurements of backscattered ultraviolet radiation: Theoretical basis. Journal of Geophysical Research: Atmospheres, 103(D14), pp.17099-17110.

Response: Thank you for your comments. After careful consideration and literature review, we have made the following explanation for your questions:

1) We strongly agree with your opinion. The aerosol index (AI), which is defined as the product of the AOD and AE (AI=AOD*AE, Nakajima et al., 2001, Liu et al., 2019). The AI allows for a more accurate quantification of aerosol column number concentration than AOD (Nakajima et al., 2001). The AI is used to indicate small particles (those that act as CCN) with a high weight (Liu et al., 2019). Our research is based on this to obtain AI with vertical structure.

Besides, we have a broad understanding of traditional AI, the AI is a way to measure how backscattered ultraviolet (UV) radiation from an atmosphere containing aerosols differs from that of a pure molecular atmosphere (Guan et al., 2010). AI is especially sensitive to the presence of UV absorbing aerosols such as smoke, mineral dust, and volcanic ash. The index value is positive when absorbing aerosols are present, whereas clouds yield nearly zero value of AI. Torres et al. (1998) and Jeong and Hsu (2008) indicated that AI varies with aerosol layer height, optical depth, and single scattering albedo. Therefore, the significance of obtaining vertical structure AI in our research content is different from that of traditional AI representation, and the specific purpose and significance will be elaborated in detail in the following text.

2) We did not use column AOD or layer AE, but layers. We apologize for not expressing this clearly in the original manuscript. We have provided specific instructions on how to calculate AI in the revised manuscript. Thank you very much for the valuable feedback.

As the innovation of this work is to obtain AI with vertical structure, which has not appeared in previous work. So all our data is based on the vertical structural distribution of altitude-latitude. That is, we use an observed relationship between CALIPSO extinction at 532nm and 1064nm to derive an altitude-latitude-based monthly climatology of Ångström exponent to compute altitude-latitude monthly climatology of vertical AI at 532nm and 1064nm between daytime and nighttime from 2007 to 2020, with vertical and horizontal resolutions of 60 m and 0.05°, respectively.

Because we focus on the characteristics of aerosols in the troposphere over the TP, we took samples from the surface at an altitude of 12km with a vertical resolution of 0.06km. We integrated the extinction coefficients of each two layers to obtain an AOD, which corresponds to the average of the AE values of each two layers. This achieves spatial matching between AOD and AE at vertical heights. In the later stage, when we used the AI obtained from MODIS for comparative testing, due to the differences in horizontal and vertical space, we chose to use the PDF and average values of AI for characterization display in order to facilitate comparison.

3) At present, according to the literature we have reviewed, the AI obtained in the ultraviolet channel can currently characterize both absorption and non-absorption aerosols. The AI obtained from our research work cannot effectively characterize the absorption and non-absorption of its aerosols, as the results we obtained are in the non-ultraviolet band range, which is also an area that we need to further explore in the future. However, the aerosol concentration represented by the vertical structure AI we obtained is not possessed by the plane information AOD. Compared to the aerosol column concentration AOD information, as AOD is an integral result of the entire layer height, it will to some extent lose some of the true changes in the vertical height of aerosols. The significance of our work is that the AI with higher reliability obtained here can more effectively obtain aerosol concentration information at the vertical height. This is the main highlight of our research work. The reason why we use AI to test the results of MODIS and ground LIDAR is to verify the effectiveness and reliability of AI. Fortunately, the test results are very consistent and reasonable. Therefore, the AI of physical meaning here which can effectively characterize aerosol concentration information at vertical heights.

Finally, after a clear definition, we will review and modify the entire text to eliminate ambiguous expressions caused by AI. Thank you again for the valuable feedback provided by the experts. We have made effective modifications to the entire text.

Liu, Z., Kar, J., Zeng, S., Tackett, J., Vaughan, M., Avery, M., Pelon, J., Getzewich, B., Lee, K.P., Magill, B., Omar, A., Lucker, P., Trepte,C., Winker, D.: Discriminating between clouds and aerosols in the caliop version 4.1 data products, Atmos. Meas. Tech., 12, 703–734, 2019.

Guan, H., Esswein, R., Lopez, J., Bergstrom, R., Warnock, A., Follette-Cook, M., Fromm, M., Iraci, L. T.: A multi-decadal history of biomass burning plume heights identified using aerosol index measurements, Atmospheric Chemistry and Physics, 10(14), 6461-6469, 2010.

Nakajima, T., Higurashi, A., Kawamoto, K., Penner, J. E.: A possible correlation between satellite - derived cloud and aerosol microphysical parameters, Geophysical Research Letters, 28 (7), 1171-1174, https://doi.org/10.1029/2000GL012186, 2001.

Torres, O., Bhartia, P. K., Herman, J. R., Ahmad, Z., and Gleason, J.: Derivation of aerosol properties from satellite measurements of backscattered ultraviolet radiation: Theoretical basis, J. Geophys. Res., 103, 17099–17110, doi:10.1029/98JD00900, 1998.

Jeong, M.-J. and Hsu, N. C.: Retrievals of aerosol single-scattering albedo and effective aerosol layer height for biomass-burning smoke: Synergy derived from "A-Train" sensors, Geophys. Res.Lett., 35, L24801, doi:10.1029/2008GL036279, 2008.

---

## Author Comment (AC2)

Dear Editor,

Thank you very much for your help with the peer review of our manuscript. We are also very grateful to all the reviewers for their help to improve the quality of our manuscript. We have made a thorough revision of this manuscript according to the reviewers' recommendations. All modified content, including added or deleted parts, in the paper are marked with color (green) in the manuscript. Please see the following pages for a point-by-point response. We hope the revised manuscript is now suitable for publication in *Earth System Science Data*.

Kind regards,
Jianping Huang

1. In eq.2, $AI_{[m,i,j]}$ and $AOD_{[m,i,j]}$ are aerosol index and aerosol optical depth, respectively; $AE_{[m,i,j]}$ is the pseudo-Ångström exponent; and [m, i, j] represent the month, latitude, and altitude respectively. Note that to match the AE, AOD is also transformed into the vertical distribution (not the column parameter) (lines 280-283). Please add the process transforming the column AOD into the vertical distribution with MODIS-AOD data.

   Response: Thank you for your suggestion. We have added the process transforming as follows in manuscript:

   The innovation of this work is to obtain AI with vertical structure, which has not appeared in previous work, and data in this manuscript are all based on the vertical structural distribution of altitude-latitude. We use an observed relationship between CALIPSO extinction at 532nm and 1064nm to derive an altitude-latitude-based monthly climatology of Ångström exponent to compute altitude-latitude monthly climatology of vertical AI at 532nm and 1064nm between daytime and nighttime from 2007 to 2020, with vertical and horizontal resolutions of 60 m and 0.05°, respectively. As we focus on the characteristics of aerosols in the troposphere over the TP, we took samples from the surface at an altitude of 12km with a vertical resolution of 0.06km. We integrated the extinction coefficients of each two layers to obtain an AOD, which corresponds to the average of the AE values of each two layers. This achieves spatial matching between AOD and AE at vertical heights. In the later stage, when using the AI obtained from MODIS for comparative testing, we used the PDF and average values of AI for characterization display in order to facilitate comparison due to the differences in horizontal and vertical space.

2. The authors use the ground-based LIDAR (Light Detection And Ranging) detection data from the hinterland of Taklimakan Desert (not from the Tibetan Plateau) to verify the validity and accuracy of the low confidence aerosol removal method and the AI calculated by CALIOP detection data. Please add the discussions on the

uncertainties in this study caused with distinct differences the Taklimakan Desert and the Tibetan Plateau.

Response: Thank you for your suggestion. We have added the discussions on the uncertainties in the revised manuscript as follows:

In general, the quality and robustness of the aerosol parameter product have improved for EC and AI with some issues that still persist in the data set which we mention below:

As we do not have ground-based LIDAR detection data on the TP, we have selected grond-based LIDAR data from the center of the Taklamakan Desert for verification and evaluation. The objectives of the verification and evaluation include the removal of low reliability aerosol targets and the validation of the effectiveness and rationality of the constructed aerosol AI parameter results. Due to the limited detection data of ground-based LIDAR, we chose a typical aerosol process detected by ground-based LIDAR (July 11, 2021), but it did not match well with the transit time and scanning area of the CALIPSO satellite, resulting in significant errors. Therefore, we choose to compare and verify the results of the average values of July in all years within the central area of the transit Taklamakan Desert detected by CALIPSO (see the green box on the left in Figure 2). Minimize spatial errors caused by significant differences in spatial positions. This kind of error is inevitable in our data processing process and will affect the consistency of detection results to some extent.

Besides, although the monthly based AI correction significantly improves the comparison between CALIPSO and MODIS, we note somewhat a larger deviation maybe occurs in winter, and the effect after correction in summer is the best and significant, which may be related to the increased probability of mistaking clouds as aerosol particles due to more convective activities in summer. This helps us to refine our research on summer aerosols over the TP.

The English language. Please make the substantial improvement on English language

and usage in the manuscript. Below I list only a part of errors:

Response: Thank you for your suggestion. We have carefully revised language issues and invited experts who are native English speakers to review and check the manuscript.

1) Line 38: more reliable the> the more reliable

Response: Thank you for your suggestion. We have corrected it.

2) Line 40: "between daytime and nighttime" should be for daytime and nighttime

Response: Thank you for your suggestion. We have corrected it.

3) Line 44: please clarify "all those facts"

Response: Thank you for your suggestion. We have corrected it.

4) Line 48: what is "aerosol troposphere"?

Response: Thank you for your suggestion. We have changed "aerosol troposphere" into "tropospheric aerosols".

5) Line 51: please modify "the recovered datasets"

Response: Thank you for your suggestion. We have corrected it.

6) Line52: "the aerosol-cloud-radiation-precipitation interaction"? There should be the aerosol-cloud interaction(ACI) aerosol-radiation interaction(ARI).

Response: Thank you for your suggestion. We have corrected it.

7) Line 100: models>modeling

Response: Thank you for your suggestion. We have corrected it.

8) Line 107-108: please modify "and its"

Response: Thank you for your suggestion. We have changed the sentence into "The primary aerosol type over the TP is dust, which is primarily contributed to the Taklimakan Desert".

9) Line 172: please correct "between our record and with different data sets"

Response: Thank you for your suggestion. We have changed "between our record and with different data sets" into "between our records and other public different data sets."

10) Line 108: spatiotemporal pattern is primarily contributed to the Taklimakan Desert".

Response: Thank you for your suggestion. We have changed the sentence into "The

primary aerosol type over the TP is dust, which is primarily contributed to the Taklimakan Desert".

---

## Referee Report (RR1)

The manuscript focuses on establishing a robust dataset of tropospheric aerosol properties over the Tibetan Plateau (TP) using CALIOP data from 2007-2020. The study provides a new vertical aerosol index (AI) parameter, extracted observations, which is comprehensively validated. The research highlights the unique climate and geographical features of the TP and the challenges in data collection for this region. The findings offer insights into the aerosol climatology of this region. I believe this paper can be published in ESSD with a few minor changes:

Minor comments:

1. Line 42 contains a repetitive phrase regarding data validation. It would be clearer if rephrased to "...which was rigorously quality-checked and validated..."

2. Line 66 needs an article for readability: "...have a crucial role..."

3. Line 118: I suggest change it to "...mainly due to the lack of a long-term ..."

4. Line 121: I suggest change it to  "...vertical dataset..."

5. Line 148: "...better lead..." to "...better leads..."

6. Additionally, a grammar check is recommended to refine the paper's language and clarity.

---

## Author Response (AR2)

Dear Editors,

Thank you very much for your help with the peer review of our manuscript. We are also very grateful to all the reviewers for their help to improve the quality of our manuscript. We have made a thorough revision of this manuscript according to the reviewers' recommendations. All modified content, including added or deleted parts, are marked with color (green) in the revised manuscript. The following are our point-by-point responses. We hope the revised manuscript is now suitable for publication in *Earth System Science Data*.

Kind regards,
Jianping Huang

The manuscript focuses on establishing a robust dataset of tropospheric aerosol properties over the Tibetan Plateau (TP) using CALIOP data from 2007-2020. The study provides a new vertical aerosol index (AI) parameter, extracted observations, which is comprehensively validated. The research highlights the unique climate and geographical features of the TP and the challenges in data collection for this region. The findings offer insights into the aerosol climatology of this region. I believe this paper can be published in ESSD with a few minor changes:

Minor comments:

1.  Line 42 contains a repetitive phrase regarding data validation. It would be clearer if rephrased to "...which was rigorously quality-checked and validated..."

Response: Thank you for your suggestion. We have corrected it.

2.  Line 66 needs an article for readability: "...have a crucial role..."

Response: Thank you for your comments. We have corrected it.

3.  Line 118: I suggest change it to "...mainly due to the lack of a long-term ..."

Response: Thank you for your suggestion. We have corrected it.

4.  Line 121: I suggest change it to "...vertical dataset..."

Response: Thank you for your suggestion. We have corrected it.

5.  Line 148: "...better lead..." to "...better leads..."

Response: Thank you for your comments. We have corrected it.

6. Additionally, a grammar check is recommended to refine the paper's language and clarity.

Response: Thank you for your comments. We have carefully reviewed all the recommendations and made corresponding revisions. In addition, we also invited experts who are native English speakers to proofread the entire text. We hope that the revised manuscript will address all your concerns.